# Screening of the Active Compounds against Neural Oxidative Damage from Ginseng Phloem Using UPLC-Q-Exactive-MS/MS Coupled with the Content-Effect Weighted Method

**DOI:** 10.3390/molecules27249061

**Published:** 2022-12-19

**Authors:** Xiao-Chen Gao, Nan-Xi Zhang, Jia-Ming Shen, Jing-Wei Lv, Kai-Yue Zhang, Yao Sun, Hang Li, Yue-Long Wang, Duan-Duan Cheng, Meng-Ya Zhao, Hui Zhang, Chun-Nan Li, Jia-Ming Sun

**Affiliations:** 1Jilin Ginseng Academy, Changchun University of Chinese Medicine, Changchun 130117, China; 2School of Chemistry and Life Sciences, Changchun University of Technology, Changchun 130012, China

**Keywords:** UPLC-Q-Exactive-MS/MS, content-effect weighting, ginseng

## Abstract

The neuroprotective properties of ginsenosides have been found to reverse the neurological damage caused by oxidation in many neurodegenerative diseases. However, the distribution of ginsenosides in different tissues of the main root, which was regarded as the primary medicinal portion in clinical practice was different, the specific parts and specific components against neural oxidative damage were not clear. The present study aims to screen and determine the potential compounds in different parts of the main root in ginseng. Comparison of the protective effects in the main root, phloem and xylem of ginseng on hydrogen peroxide-induced cell death of SH-SY5Y neurons was investigated. UPLC-Q-Exactive-MS/MS was used to quickly and comprehensively characterize the chemical compositions of the active parts. Network pharmacology combined with a molecular docking approach was employed to virtually screen for disease-related targets and potential active compounds. By comparing the changes before and after Content-Effect weighting, the compounds with stronger anti-nerve oxidative damage activity were screened out more accurately. Finally, the activity of the selected monomer components was verified. The results suggested that the phloem of ginseng was the most effective part. There were 19 effective compounds and 14 core targets, and enriched signaling pathway and biological functions were predicted. After Content-Effect weighting, compounds Ginsenosides F1, Ginsenosides Rf, Ginsenosides Rg_1_ and Ginsenosides Rd were screened out as potential active compounds against neural oxidative damage. The activity verification study indicated that all four predicted ginsenosides were effective in protecting SH-SY5Y cells from oxidative injury. The four compounds can be further investigated as potential lead compounds for neurodegenerative diseases. This also provides a combined virtual and practical method for the simple and rapid screening of active ingredients in natural products.

## 1. Introduction

Oxidative damage to nerves has been implicated in the pathogenesis of various chronic neurodegenerative diseases, such as Alzheimer’s disease and Parkinson’s disease [1]. Neuropathological features include oxidative damage, neuronal and synaptic loss [2]. Proteins, deoxynucleic acids and lipid membranes can be damaged by oxidative damage caused by reactive oxygen species generated after an oxidative burst or the presence of excess free transition metals, thereby, disrupting the cellular function and integrity. 

The presence of H_2_O_2_ causes lipid peroxidation and DNA damage, which results in apoptosis in a variety of cell types [3,4,5]. Recently, therapeutic strategies to prevent or delay reactive-oxygen-species-induced apoptosis have been proposed as a viable option to treat these diseases [6,7,8]. Many synthetic chemicals, such as phenolic compounds have been proven to be strong radical scavengers; however, they usually have some severe adverse effects [9]. Therefore, research attention has focused on searching for natural substances with neuroprotective potential to protect from neural oxidative damage.

Regarding ginseng, the root and rhizome of *Panax ginseng* C.A.Mey, studies have shown wide medical benefits, such as curing central nervous system disease, endocrine system disease, cardiovascular system disease and so on [10,11,12,13,14]. Research reported neuroprotective value for central nervous system disorders and neuronal diseases attributed to the ginsenosides, which are the main bioactive compounds in Ginseng [15,16]. Studies reported neuroprotective value for central nervous system disorders and neuronal diseases attributed to the ginsenosides, which are the main bioactive compounds in Ginseng—in total, over 150 ginsenosides have been identified thus far [17,18,19]. 

In addition to whole organs, some studies have focused on the accumulation and histologic distribution patterns of ginsenosides in different parts of the main root, which is considered as the major medicinal part. In particular, studies have shown that xylem and phloem belong to the vascular system and express multiple genes involved in the synthesis of ginsenosides [14,20,21,22,23,24]. A ginsenoside was synthesized in vascular tissue, such as the phloem, and then transported to the roots’ periderm and thin walls with transporters acting as defenders against insects and animals [13,25].

In order to study the material basis and action mechanism of compound compatibility, network pharmacology was widely used. It uses the network model to express and study on the interaction amongst the active component, target and disease, whose behaviors were further elucidated by a molecular docking simulation intended to simulate the docking of small-molecule ligands and large-molecule proteins, and the docking results were evaluated using the binding energy. Through calculation, active compounds and key target proteins for the treatment of diseases can be screened [26,27,28].

Despite the convenience, this is not without its shortcomings—for example, each component was treated equally, which is unreasonable, and it ignores the effect of the content in each component [29]. Specifically, it relies too much on public databases to accurately reflect the presence and proportion of compounds in medicinal materials, as the active ingredients in the database are not necessarily present in the material under investigation, or the database may not be complete. Thus, it was necessary to verify and supplement the information of the database through actual measurements using by UPLC-Q-Exactive-MS/MS or other methods. 

Second, most of the studies in network pharmacology were based on qualitative “component-target-disease” analysis, and using the content as a key factor can prevent a very low level of ingredients from being defined as a critical ingredient. Therefore, it was necessary to introduce the content coefficient, which can more accurately locate the components with high content to the center of the network and transform one-dimensional qualitative research into two-dimensional qualitative and quantitative research. 

Moreover, the traditional method only describes the interaction between the active compound and the target but does not mention the strength of the interaction and cannot more accurately predict the target with a stronger effect and its related pathways, which affect the analysis of its mechanism of action. Similarly, it was necessary to introduce the effect coefficient, which can more accurately capture the more important compounds and the targets playing the main role in the network, which can combine and have a certain intensity.

Some factors of traditional network pharmacology will lead to problems in the completeness and accuracy of prediction data, and thus there is an urgent need to design values about the content and effect and integrate them into the corresponding network algorithm to evaluate the effects. Inspired by the above applications, an effective strategy was used on UPLC-Q-Exactive-MS/MS with the Content-Effect weighted method to screen the active compounds in ginseng for neural oxidative damage.

## 2. Results

### 2.1. The Protective Effects of Different Parts of Ginseng on H_2_O_2_ Damage to SH-SY5Y Cells

All the different parts in ginseng showed a certain protective effect on H_2_O_2_ damage to SH-SY5Y cells. Microscopic observation showed that good cell apposition, adequate extension, good refractive index, clear medium without impurities and a large number of dividing cells were visible in the control group. In the model group, the growth state of the adherent cells was poor and not clarified, while, with the prolongation of the drug action time, the cells showed a distinct growth promotion state: the number of adherent cells gradually increased. MTT experiments showed that, when the concentrations were 25, 50 and 100 μg/mL, there was a clear dose-dependent effect. The vitality gradually became stronger, and the most significant reduction in damage was observed in the phloem group (Figure 1).

### 2.2. Identification of Compounds in Phloem of the Ginseng by UPLC-Q-Exactive-MS/MS

The samples of the phloem group were analyzed by UPLC-Q-Exactive-MS/MS, and the total ion flow diagram (negative ion mode) is shown in Figure 2. The identification of 19 ginsenoside compounds based on retention time, molecular ion peaks, related ion fragments and related literature mass spectrometry data (Table 1).

### 2.3. Analysis of the Network Pharmacology

The target information of 19 ginsenosides was collected by TCMSP, a systematic pharmacological analysis platform for traditional Chinese medicine, and 439 relevant targets were obtained by removing duplicate items. The target information was collected and screened using the Gene Cards database with “neural oxidative damage” and “Oxidative nerve injury” as the keywords. The disease targets with Score > 1 were selected, and 6197 targets related to memory enhancement were obtained by removing duplicates. After mapping, 400 disease effect targets were obtained.

The 400 effect targets were entered into the STRING database, and the minimum required interaction score was set to 0.9 to hide the nodes disconnected from the network to obtain the protein–protein interaction network. The results of the above analysis were imported into Cytoscape 3.7.2 software in TSV format, and 14 core targets were screened by (degree > mean× 2) to construct a metabolite-core target network by Cytoscape (Figure 3).

KEGG pathway enrichment and GO analysis were performed for 14 core targets using the DAVID database. KEGG pathway enrichment showed 119 signaling pathways, and the top 20 pathways according to *p*-value are shown in Figure 4. GO analysis revealed 221 enriched processes, including 175 biological processes, 24 molecular functions and 22 cellular components. The top 30 ranked according to *p*-value are shown in Figure 5.

### 2.4. Analysis of the Molecular Docking

The binding energy results of molecular docking between the 19 identified ginsenosides and the 14 core targets are shown in Figure 6. Figure 7 shows the results of the highest binding energy of ginsenosides docked with each core target. It is generally believed that, when the binding energy is less than 0, this indicates that the component and the target can be bound together. The lower the binding energy, the stronger the binding ability of the two and the more stable the conformation formed. 

### 2.5. Analysis of Content-Effect Weighted Method

Based on the content of components and the binding energy of molecular docking, 19 ginsenosides were weighted. Ginsenoside F1, Ginsenoside Rd, Ginsenoside Rg_1_ and Ginsenoside Rf were screened as important compounds with functions against neural oxidative damage, and the results are shown in Table 2.

### 2.6. The Protective Effect of Active Compounds on H_2_O_2_-Induced SH-SY5Y Cells

The protection of active compounds against H_2_O_2_-induced SH-SY5Y cells is shown in Figure 8. The results showed that the viability of H_2_O_2_-induced SH-SY5Y cells significantly decreased in the model group. Compared with the blank group, the cell viability of H_2_O_2_-induced SH-SY5Y cells was increased by the intervention of the five active compounds compared with the model group, and there were significant differences. Thus, all five active compounds were effective in protecting H_2_O_2_ damage to SH-SY5Y cells with the relative protective order of Ginsenoside Rf > Ginsenoside F1 > Ginsenoside Rk_1_ > Ginsenoside Rg_1_ > Ginsenoside Rd. (*p* < 0.05 or *p* < 0.01).

## 3. Discussion

Traditional network pharmacology was used to predict the compounds of ginseng phloem with protective effect on neural oxidative damage, including Ginsenoside Rd, Ginsenoside Rk_1_, Ginsenoside Rg_1_ (with the high original degree value). However, this might not be accurate, because it treated each component equally, ignoring the impact of the content. The content coefficient and the effect coefficient were introduced by the weighting method to establish the relationship between the content and the effect, and thus the active ingredient could be predicted more accurately.

The order of activity strength predicted by Content-Effect weighted network pharmacology was Ginsenoside F1, Ginsenoside Rd, Ginsenoside Rf and Ginsenoside Rg_1_. This ranking was clearly different from the unweighted ones; thus, it was necessary to verify the prediction through monomeric activity experiments, in which the top four weighted compounds and the second top unweighted compound Ginsenoside Rk_1_ (with the original degree value 9) were used. The first and the third top unweighed compounds, Ginsenoside Rd (with the original degree value 10) and Ginsenoside Rg_1_ (with the original degree value 9), were excluded because they were already included in the top four weighed counterparts. 

The results showed that all the above five compounds were protective against neural oxidative damage, which was consistent with the conclusions reported in the literature, in which Ginsenoside Rf and Ginsenoside F1 were more prominent. In the results of the unweighted network pharmacology, Ginsenoside Rf and Ginsenoside F1 ranked seventh and eighth; however, the weighted prediction ranked them third and first. Therefore, this order was significantly improved, which is consistent with the results of the monomer activity validation. 

The relative contents of these two compounds were 10.11% and 22.04%, Ginsenoside F1 was the ginsenoside with the highest content in the phloem of ginseng; thus, it was reasonable to be one of the strongest neuroprotective activity compounds. Similarly, Ginsenoside Rk_1_ had its predicted activity ranked second in the unweighted network pharmacology; however, its content was only 0.43%, located in the lowest content of the 19 compounds. The monomer activity was not particularly significant strong in the five ginsenosides; therefore, it was reasonable to predict that its activity ranked 15th after weighting.

All the above results showed that Content-Effect weighted network pharmacology transformed the traditional 1D (one-dimensional) qualitative parameter into 2D qualitative and quantitative parameters, thereby, improving the accuracy of the data with reasonable, effective and necessary changes. This could more accurately predict the activity of compounds compared to traditional network pharmacology. Based on the above research, an activity screening method was established, which was beneficial to the activity screening of other natural products.

## 4. Materials and Methods

### 4.1. Materials and Chemicals

Fresh roots of 5-year-old ginseng were collected from Fusong county, Jilin province, China. The samples were taxonomically identified by Changchun University of Chinese Medicine, and a voucher specimen (No. 202105) was deposited at the laboratory of Jilin Ginseng Academy, Changchun University of Chinese Medicine, Changchun, China.

UPLC grade acetonitrile was obtained from Tedia Company Inc., (Fairfield, OH, USA). Purified water was made by a water purifier (Global Water Solution Ltd., Randolph, MA, USA). Other reagents and chemicals of analytical grade, including methanol, ethanol, n-butanol, trichloromethane, DMSO, Na_2_HPO_4_·12H_2_O and NaH_2_PO_4_·2H_2_O, were purchased from Beijing Chemical Works, (Beijing, China). Phosphate buffer (PBS, 0.1M, pH 7.6), containing 0.05% 3-(trimethylsilyl)-propionic-2,2,3,3-d4 acid sodium salt (TSP) as an internal standard, was acquired from Cambridge Isotope Laboratories Inc. (Andover, MA, USA). Deuterium Oxide (D_2_O 99.9% atom% D) was purchased from Tenglong Weibo Technology Co., Ltd. (Qingdao, China). The standards Ginsenoside Rg_1_, Ginsenoside Rf, Ginsenoside F1, Ginsenoside RK_1_, Ginsenoside Rd were all obtained from Beijing Science and Technology (Beijing, China), the purity of all standards was greater than 98%.

### 4.2. Sample Preparation

The whole fresh plant was thoroughly rinsed with deionized water, the main root was cut down, and two parts in the cross section of main root—namely, the xylem and phloem—were peeled off. Then, all the cut samples were dried in an oven at 50 °C for 72 h and finally smashed into a powder sieved through a 20-mesh. A 2.0 g pulverized sample was weighed into a 50 mL centrifuge tube, and 40 mL of water-saturated n-butanol was added. The sample was sonicated for 30 min, filtered, evaporated until dryness on a water bath at 70 °C, stored in a desiccator and filtered with a 0.22 µm filter membrane for later use. Extracts from three different parts of ginseng, the main root, phloem and xylem, with five of each kind, for a total of 15 ginseng samples were prepared according to the above mentioned extraction method for testing.

### 4.3. Cell Culture

The human neuroblastoma SH-SY5Y cells were cultured in MEM medium and 50% Ham’s F-12 containing 10 fetal calf serum, 100 U/mL penicillin and 100 U/mL streptomycin in a humid atmosphere of 95% air and 5% CO_2_. SH-SY5Y cells were plated on plates pretreated with DMSO (1/1000) and various concentrations (1, 10 and 100 μM) of extracts from three different parts of ginseng for 24 h, the samples were exposed to 150 µM of H_2_O_2_ for 24 h at the same concentrations. A 30% stock solution of H_2_O_2_ was freshly prepared for each experiment to produce oxidative stress. The control cells were added the same medium without H_2_O_2_ and extract from three different parts of ginseng. A hemocytometer was used to count and differentiate viable and dead cells by adding 10% Trypan Blue.

### 4.4. MTT Assay

SH-SY5Y cells were plated at a density of 1 × 10^4^ cells per well in 96-well plates, and the cell viability was determined using the conventional MTT reduction assay. Briefly, after 24 h exposure to H_2_O_2_, 4 0 μL of MTT (2 mg/mL in PBS) was added to each well, and the cells were incubated at 37 °C for 4 h. The supernatants were aspirated carefully, and 100 μL of dimethyl sulfoxide (DMSO) was added to each well to dissolve the precipitate. The absorbance at 570 nm was measured with a microplate reader (BIO-RAD Model 3550, Hercules, CA, USA).

### 4.5. UPLC-Q-Exactive-MS/MS Conditions

Chromatographic conditions: an Agilent SB-C18 chromatographic column (4.6 mm × 100 mm, 1.8 microns, P.N. 828975-902, S.N. USWFM02237, Agilent Technologies, Inc., Palo Alto, CA, USA) was used with the following gradient elution: mobile phase: 0.1% formic acid water (B) and acetonitrile (C). The optimal elution conditions were as follows: 0–10 min: 5% C, 10–15 min: 35% C, 15–30 min: 40% C, 30–35 min: 50% C, 35–50 min: 100% C; flow rate: 0.3 mL/min; injection volume: 10 μL; and column temperature 30 °C. Mass spectrometry conditions: an Agilent 1100 UPLC/MSD Trap mass spectrometer 6320 (Agilent) equipped with an electrospray ionization source was used in both positive and negative ion mode. Electrospray ion source (ESI), positive and negative ion mode detection, drying gas temperature: 350 °C, sheath gas flow rate: 4 × 10^6^ Pa, auxiliary gas flow rate: 1 × 10^6^ Pa, auxiliary gas temperature: 300 °C, scanning mode: Full scan-ddMS2, resolution: 70,000 FWHM and mass scanning range: 150–2000 *m*/*z*.

### 4.6. Network Pharmacology Analysis

#### 4.6.1. Construction of the Compound-Target and Disease-Target Networks

Based on the structure of active components identified by UPLC-QTOF-MS/MS, Pubchem and Swiss Target Prediction predicted component targets, a gene card database was used to identify known targets associated with oxidative nerve injury. Ginsenosides were derived as neuroprotective effect targets by mapping component targets to disease targets.

#### 4.6.2. Protein–Protein Interaction and Pathway-Enrichment Analysis

Topological analysis of the component-target network was done using a network analyzer. The effector targets were placed in the STRING (https://string-db.org/, Version 11.5, accessed on 9 July 2022.) [30]. In order to obtain core targets, nodes above the mean of 4 degrees were identified as interacting proteins. The PPI network, ginseng differential metabolite-core target network was visualized and analyzed using Cytoscape.

The Metascape database was used for functional enrichment analysis, the core targets were entered into the Metascape database, the species was selected as H. sapiens, the PValue Cutoff was set to 0.01, and the remainder were kept as default settings [31,32]. GO contains biological process (BP), molecular function (MF) and cellular composition (CC), and GO analysis and KEGG-related pathway analyses were performed using the Metascape database (*p* < 0.05) [33].

#### 4.6.3. Correlation Analysis

To examine the correlation between biomarkers of ginseng phloem, Pearson correlation analysis of chemical markers in TR, LR and ZR was performed with SPSS 22.0 and visualized using heatmaps. 

### 4.7. Molecular Docking

Small molecule ligands and protein receptors were processed before docking, and ligands and non-protein molecules were removed from the protein using PyMol (Version 2.5.1; The PyMOL Molecular Graphics System, Schrödinger, LLC, New York, NY, USA) and saved in pdb format. We converted the molecules from mol2 format to pdb format to save them. We opened all molecule files in AutoDock Tools (Version 1.5.6; Department of Molecular Biology, The Scripps Research Institute, La Jolla, CA, USA), hydrogenated and charged the molecules separately and saved them as pdbqt files. We opened all proteins, hydrogenated, charged, added the protein type, etc. and saved them in pdbqt format as well.

We imported the processed small molecule ligands and protein receptor structures, the Grid Box coordinates and box size were set, and the calculations were run using the “local search” algorithm with default parameters. The docking results were evaluated by the binding energy value. A binding energy value less than 0 indicates that the ligand and the receptor could bind spontaneously, and a binding energy −5.0 kJ/ mol indicates that the ligand binds well to the receptor. The conformation with the lowest binding energy was selected and displayed on a graph using PyMol (Version 2.5.1).

### 4.8. Establishment of the Weighted Value

Ginsenosides contain a variety of active components, and the relative contents of different medicinal parts were also different. The molecular docking binding energy was an important parameter to measure the binding degree between active substances and target proteins.

In order to better explain the importance of each component content in the biological effect, the ginsenoside content and molecular docking binding energy were linked, and the relationship was established by weighting. The content coefficient, efficiency coefficient and weighted weight were introduced to measure the importance of ginsenosides in the anti-neural oxidative damage weighted network, which could be expressed by the following formula:(1)$$C=\frac{A}{A_{\min}}$$
where C represents the content coefficient, A represents the peak area of each component, and $A_{\min}\;$represents the peak area representing the smallest component in content.
(2)$$E=\frac{\sum B_{i}}{B_{\max}}$$
where B represents the binding energy of a compound to the target protein, $B_{i}$ represents the binding energy of a compound to different binding target proteins, and $B_{\max}$ represents the maximum value of the sum of the binding energy of a compound with all binding target proteins.
(3)$$W_{d}=C\cdot E\cdot D$$
where $W_{d}\;$represents the weighted value, and D represents the degree of binding target proteins.

According to the order of weighted value, active compounds were selected.

## 5. Conclusions 

In this study, four active compounds with protective effects against neural oxidative injury were screened as leading compounds for the treatment of neurodegenerative diseases. A more accurate screening method was established to facilitate the active screening of natural products.

## Figures and Tables

**Figure 1 molecules-27-09061-f001:**
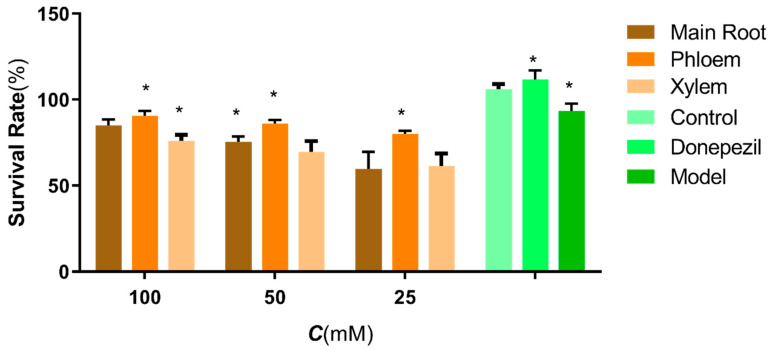
Effects of the different parts in ginseng on H_2_O_2_ damage to SH-SY5Y cells. The data are presented as the means ±SEM (n = 3). *: *p* < 0.05 compared with the model group.

**Figure 2 molecules-27-09061-f002:**
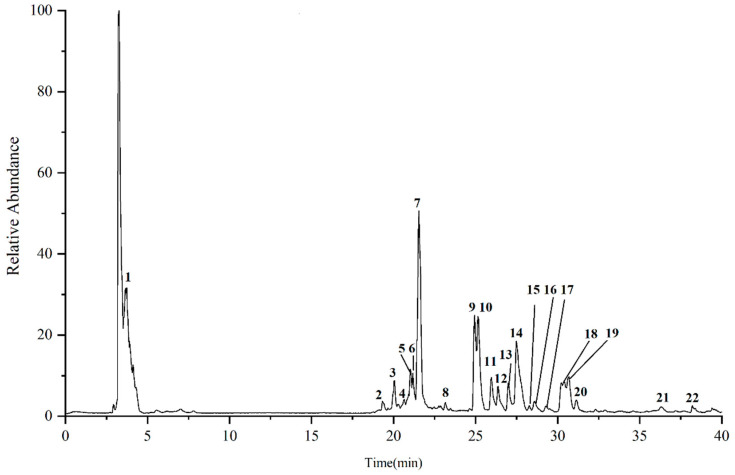
Total ion flow diagram (negative ion mode) of phloem in ginseng.

**Figure 3 molecules-27-09061-f003:**
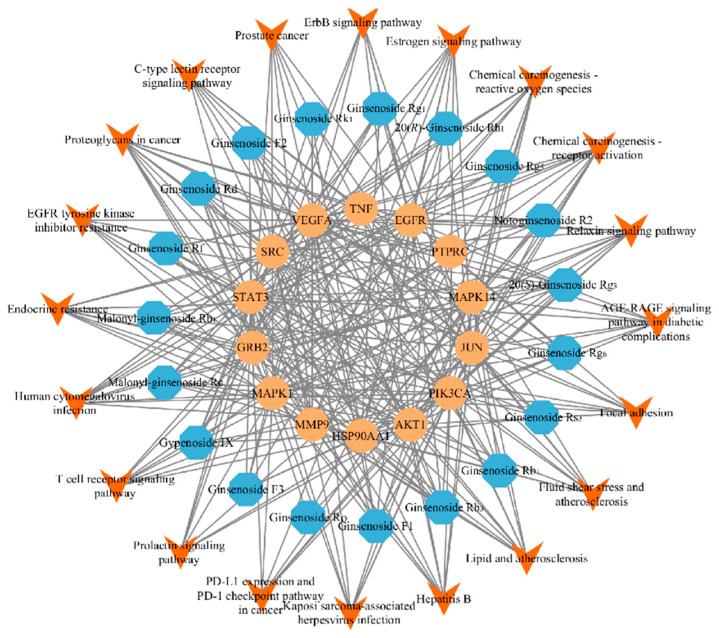
Protein–protein interaction network.

**Figure 4 molecules-27-09061-f004:**
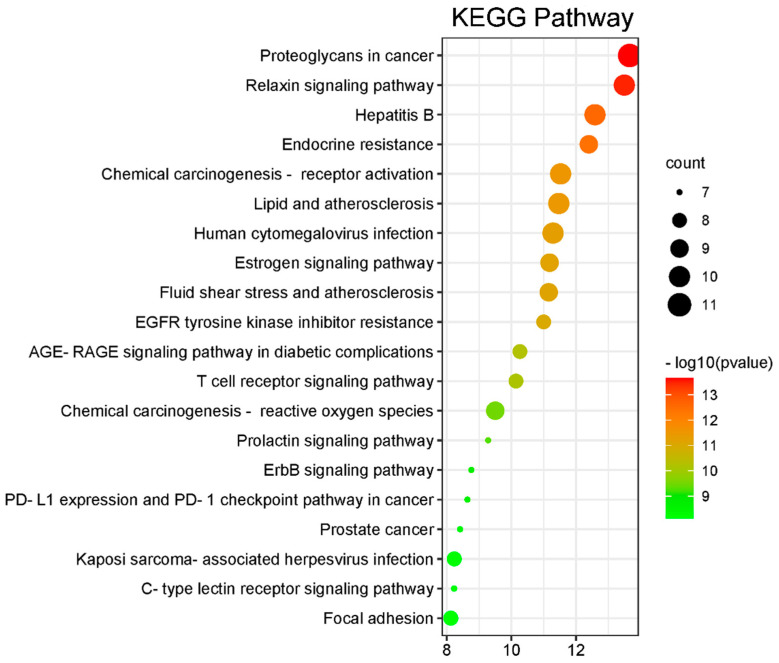
KEGG pathway enrichment.

**Figure 5 molecules-27-09061-f005:**
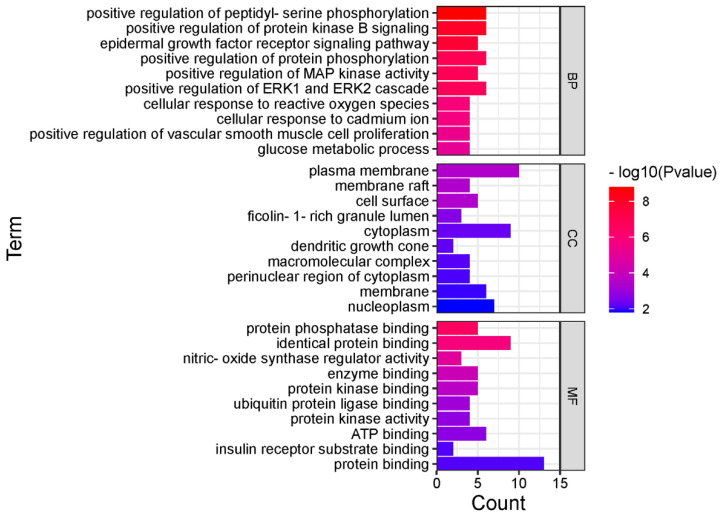
GO analysis.

**Figure 6 molecules-27-09061-f006:**
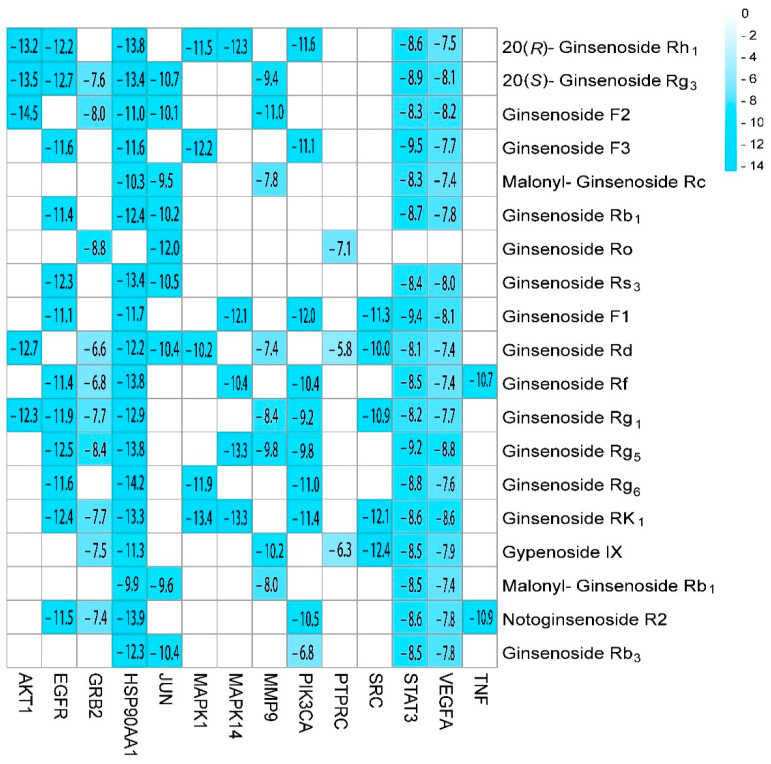
The binding energy of molecular docking between the nineteen identified ginsenosides and the fourteen core targets.

**Figure 7 molecules-27-09061-f007:**
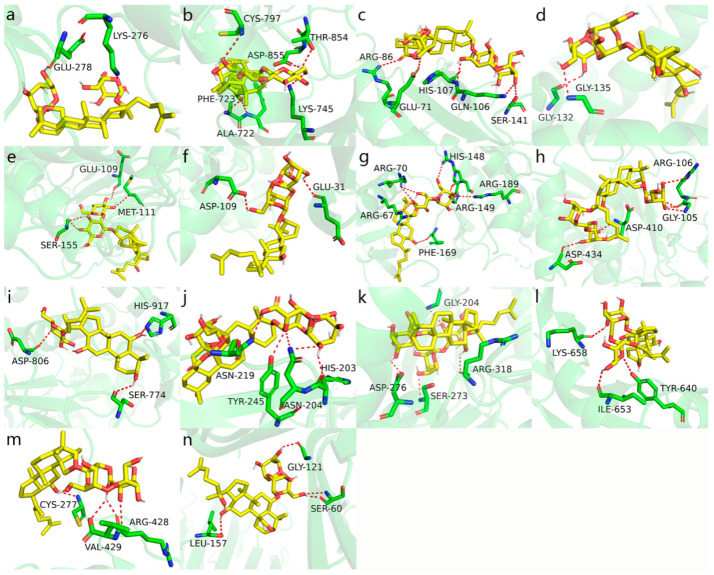
The highest binding energy of ginsenosides docked with each core target. (**a**) AKT1 docking with Ginsenoside F2; the binding energy was −14.5. (**b**) EGFR docking with 20(*S*)-Ginsenoside Rg_3_; the binding energy was −12.7. (**c**) GRB2 docking with Ginsenoside Ro; the binding energy was −8.8. (**d**) HSP90AA1 docking with Ginsenoside Rf; the binding energy was −13.8. (**e**) JUN docking with 20(*S*)-Ginsenoside Rg_3_; the binding energy was −10.7. (**f**) MAPK1 docking with Ginsenoside Rk_1_; the binding energy was −13.4. (**g**) MAPK14 docking with Ginsenoside Rk_1_; the binding energy was −13.3. (**h**) MMP9 docking with Gypenoside IX; the binding energy was −10.2. (**i**) PIK3CA docking with Ginsenoside F1; the binding energy was −12. (**j**) PTPRC docking with Ginsenoside Ro; the binding energy was −7.1. (**k**) SRC docking with Gypenoside IX; the binding energy was −12.4. (**l**) STAT3 docking with Ginsenoside F3;the binding energy was −9.5. (**m**) VEGFA docking with Ginsenoside Rg_5_; the binding energy was −8.8. (**n**) TNF docking with Ginsenoside Rf; the binding energy was −10.7.

**Figure 8 molecules-27-09061-f008:**
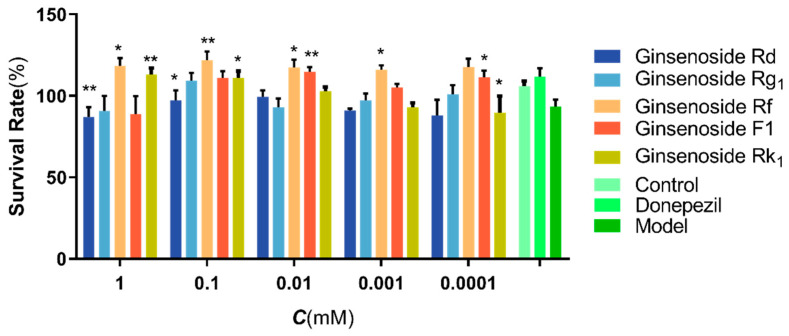
Effects of five active compounds in SH-SY5Y cells induced by H_2_O_2_. The data are presented as the means ± SEM (n = 3) *: *p* < 0.05 compared with the model group, **: *p* < 0.01 compared with the model group.

**Table 1 molecules-27-09061-t001:** Summary of compounds of phloem in ginseng identified by UPLC-Q-Exactive MS/MS.

No.	RT/min	[M-H]-	[M+HCOO]-	MS/MS Fragment Ions	Peak Area	Compound
1	4.85	-	683.5000	475.3746; 161.0438	7,483,087.36	Ginsenoside F1
2	19.32	851.4600	-	-	382,261.81	Unknown
3	20.06	799.0000	-	637.4302; 475.3775; 161.0445	2,462,891.31	Ginsenoside Rg_1_
4	20.19	769.4744	-	637.4314; 475.3788	399,896.95	Notoginsenoside R2
5	21.01	1163.5860	-	1077.5839; 945.5390; 783.4875	521,391.11	Malonyl-ginsenoside Rc
6	21.18	1193.5960	-	945.5067; 783.4598; 621.4125	417,224.70	Malonyl-ginsenoside Rb_1_
7	21.54	-	991.9580	783.4615; 621.4111	3,865,902.28	Ginsenoside Rd
8	23.15	1077.5851	-	945.5088; 621.4125	151,726.43	Ginsenoside Rb_3_
9	24.91	955.0521	-	631.3852; 455.3498; 119.0336	2,640,554.75	Ginsenoside Ro
10	25.17	799.4849	-	475.3657; 221.0609; 101.0247	3,432,721.17	Ginsenoside Rf
11	25.96	716.3397	-	783.4935; 621.4398; 459.3845	1,542,993.88	Gypenoside IX
12	26.35	-	769.3937	769.4711; 637.4299; 475.3777	1,580,772.42	Ginsenoside F3
13	26.98	-	829.7129	621.4223; 459.3791; 161.0381	1,869,503.76	Ginsenoside F2
14	27.48	1107.1880		945.5067; 783.4598; 621.4125	3,839,623.34	Ginsenoside Rb_1_
15	28.27	597.3012	-	-	149,999.06	Unknown
16	28.57	-	683.3636	475.3784; 161.0442	275,862.57	20(*R*)-Ginsenoside Rh_1_
17	29.29	-	811.6000	221.0579; 161.0381; 101.0186	146,976.50	Ginsenoside Rk_1_
18	30.23	991.5514	-	-	998,851.03	Unknown
19	30.69	-	811.1000	603.4211; 441.3664; 221.0579	1,155,965.76	Ginsenoside Rg_5_
20	31.12	783.4900	-	621.4309; 459.3768; 221.0598	296,078.61	20(*S*)-Ginsenoside Rg_3_
21	29.79	-	825.4038	783.4897; 621.4352; 459.3832	157,781.60	Ginsenoside Rs_3_
22	35.46	-	811.2000	619.4207; 457.3665	169,659.90	Ginsenoside Rg_6_

**Table 2 molecules-27-09061-t002:** Weighted values of nineteen ginsenosides using the Content-Effect weighted method.

NO.	Compound	Original Degree	Original Binding Energy	Content Coefficient	Effect Coefficient	Weighted Value	Original * Order	Order after Weighted	Order Changing after Weighted
1	Ginsenoside Rd	10	90.8	26.30	3.25	856.02	1	2	↓1
2	Ginsenoside Rk_1_	9	100.8	1.00	3.61	32.52	2	13	↓11
3	Ginsenoside Rg_1_	9	89.2	16.76	3.20	482.69	3	4	↓1
4	20(*R*)-Ginsenoside Rh_1_	8	90.7	2.01	3.24	52.35	4	11	↓7
5	Ginsenoside Rg_5_	8	85.6	7.86	3.07	193.04	5	7	↓2
6	20(*S*)-Ginsenoside Rg_3_	8	84.3	1.03	3.02	24.95	6	15	↓9
7	Ginsenoside Rf	8	79.4	23.36	2.85	531.74	7	3	↑4
8	Ginsenoside F1	7	75.7	50.91	2.71	966.99	8	1	↑7
9	Ginsenoside F2	7	71.1	12.72	2.55	226.90	9	6	↑3
10	Notoginsenoside R2	7	70.6	2.72	2.53	48.19	10	12	↓2
11	Gypenoside IX	7	64.1	10.50	2.30	169.02	11	8	↑3
12	Ginsenoside Rg_6_	6	65.1	1.15	2.33	16.16	12	18	↓6
13	Ginsenoside F3	6	63.7	10.76	2.28	147.34	13	9	↑4
14	Ginsenoside Rs_3_	5	52.6	1.07	1.89	10.12	14	19	↓5
15	Ginsenoside Rb_1_	5	50.5	26.12	1.81	236.43	15	5	↑10
16	Ginsenoside Rb_3_	5	45.8	2.01	2.46	24.78	16	16	-
17	Malonyl-ginsenoside Rb_1_	5	43.4	2.84	1.56	22.08	17	17	-
18	Malonyl-ginsenoside Rc	5	43.3	3.55	1.55	27.49	18	14	↑4
19	Ginsenoside Ro	3	27.9	17.97	1.00	53.90	19	10	↑9

* The original order was sequenced according to the numerical value of the original degree with the same, it was sequenced according to the absolute value of the sum of the original binding energy.

## Data Availability

The study did not report any data.

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
