# Peer review of "Screening of the Active Compounds against Neural Oxidative Damage from Ginseng Phloem Using UPLC-Q-Exactive-MS/MS Coupled with the Content-Effect Weighted Method"

_molecules, 2022, doi:10.3390/molecules27249061_

Round 1

Reviewer 1 Report

The authors need address these issues before further consideration。

1. In figure 1, the data of model group treated with vehicle was only presented as average data (100%), no SD data was shown.

2. Only 19 ginsenosides were identified by using UPLC-QE-MS/MS, which indicated that the MS data was not well analyzed by the authors. The authors need re-analyze the MS data.

3. Whether the 1mM selected 5 ginsenosides could affect the SH-SY5Y cell survival? Also the authors need give more detailed method to explain how could these ginsnenosides maintain a well solubility in cell culture medium.

4. The contents of these 19 ginsenosides were determined using UPLC-QE-MS/MS? Any internal standard was used?  

Author Response

Dear Reviewer, 
Thank you for your comments concerning our manuscript entitled. We have studied your 
comments carefully and have made correction which we hope meet with your approval. We are 
sending the revised manuscript to you and the revised portions are in red. We made further 
changes and explanations as follow.
1. In figure 1, the data of model group treated with vehicle was only presented as average data 
(100%), no SD data was shown.
Response 1: This error has been modified.
2. Only 19 ginsenosides were identified by using UPLC-QE-MS/MS, which indicated that the MS 
data was not well analyzed by the authors. The authors need re-analyze the MS data.
Response 2: Due to the author's negligence, there were some errors in the coding of the common 
peaks and two numbers of peeks were lost. The data had been re-analyzed.
3. Whether the 1mM selected 5 ginsenosides could affect the SH-SY5Y cell survival? Also the 
authors need give more detailed method to explain how could these ginsnenosides maintain a well 
solubility in cell culture medium.
Response 3: According to the author's previous research [1], 1mM of ginsenosides was used was 
used in activity research. At this concentration, three compounds, Ginsenoside Rf, Ginsenoside Rk1 
and Ginsenoside Rd, had protective effects on H2O2-induced damage to SH-SY5Y cells.
Due to the author's carelessness, it was forgot to describe that DMSO (1/1000) was added to 
dissolve ginsenosides. 
4. The contents of these 19 ginsenosides were determined using UPLC-QE-MS/MS? Any internal 
standard was used? 
Response 3: Internal standard was not used. Under the fixed system, the ionization intensity of 
similar compounds was basically the same, the relative content, peak area, was used to represent 
the content.
[1]杨银平,律广富,张乔,姚梦杰,梁新合,李茎,张辉,孙佳明.基于细胞代谢组学技术的人参皂苷 Rb1
对 SH-SY5Y 细胞保护机制研究[J].分析化学,2019,47(01):49-58.DOI:10.19756/j.issn.0253-
3820.181149.
[1]YANG, LYU, ZHANG, YAO, LIANG, LI, ZHANG *, SUN * Metabolomics Study of Protection 
Mechanism of GinsenosideRb1on Injury of SH-SY5Y Cell Induced by Glutamic Acid [J].
Chinese Journal of Analytical Chemistry, 2019,47(01):49-58. DOI:10.19756/j.issn.0253-
3820.181149

Reviewer 2 Report

In this study, the potential protective effects of the different root parts of ginseng on hydrogen peroxide-induced cell death of SH-SY5Y neurons were investigated. Then, UPLC-Q-Exactive-MS/MS approach was used for the characterization of chemical compositions from the active function. In addition, network pharmacology considering "content-effect" weighting was screened for the potential active compounds. The whole manuscript was complete, whereas some suggestions might be considered.

1.     The language of manuscript could not match the standard of this journal, in which the tense in different sentences should be consistent, and some words were colloquial.

2.     Many major components have not been identified in the active function, are these components involved in activities?

3.     Only one minor compound was selected for the active test, which is inconclusive.

4.     The format of title in “References” should be modified. The initials should be changed to lowercase except for the first word.

5.     Line 15: “ne urologic” => “neurologic”

6.     Line 29: “sand” => “and”

7.     Line 52: Comma should be modified to English format.

8.     Line 69: “Which behaviors are further elucidated by a molecular docking simulation intending to simulate the docking of small molecule ligands and large-molecule proteins”, this sentence presented the language disorder.

9.     Line 96: “Inspired by the above applications, an effective strategy used on UPLC-Q-Exactive-MS/MS with “Content-Effect” weighted method to screen the active compounds against neural oxidative damage in ginseng”, this sentence presented the language disorder.

10.   Line 108: “MTT experiments showed when the concentration was 25, 50, and 100 μg/mL, there was an obvious dose-dependent effect”, this sentence presented the language disorder.

11.   Figure 2: Peaks 13 and 14 disappeared.

12.   Table 1: compound 6 20(S)-Ginsenoside Rg3 => 20(S)-Ginsenoside Rg3

13.   Table 1: compound 14 20(R)-Ginsenoside Rh1 => 20(S)-Ginsenoside Rh1

14.   Line 164: “can bind together” => “can be bound together”

15.   Figure 6: The name of compounds in this figure should be modified more standard.

16.   Table 2: The format of the compounds’ name should be unified with Table 1.

17.   Line 195-196 “was significantly decreased” => “significantly decreased”

18.   “Discussion” section: “Rk1” => “Rk1

                   “Rg1” => “Rg1

                   “20(R)-Ginsenoside Rh1” => “20(R)-Ginsenoside Rh1

19.   Line 231 “six” => “five”

20.   Line 270 “CO2” => “CO2” 

        “H2O2” => “H2O2

21.   Line 278 “1 × 104” => “1 × 104

22.   Line 291, Line 295, Line 296 “30℃” => “30ºC”

23.   Line 295: “4×106” => “4×106

24.   Line 297: “m/z” => “m/z
